# Enhanced Activity of NLRP3 Inflammasome in the Lung of Patients with Anti-Synthetase Syndrome

**DOI:** 10.3390/cells12010060

**Published:** 2022-12-23

**Authors:** Espiridión Ramos-Martinez, Angel E. Vega-Sánchez, Gloria Pérez-Rubio, Mayra Mejia, Ivette Buendía-Roldán, Montserrat I. González-Pérez, Heidegger N. Mateos-Toledo, Warrison A. Andrade, Ramcés Falfán-Valencia, Jorge Rojas-Serrano

**Affiliations:** 1Experimental Medicine Research Unit, Facultad de Medicina, Universidad Nacional Autónoma de México, Mexico City 06720, Mexico; 2Interstitial Lung Disease and Rheumatology Unit, Instituto Nacional de Enfermedades Respiratorias Ismael Cosío Villegas, Mexico City 14080, Mexico; 3HLA Laboratory, Instituto Nacional de Enfermedades Respiratorias Ismael Cosío Villegas, Mexico City 14080, Mexico; 4Translational Research Laboratory on Aging and Pulmonary Fibrosis, Instituto Nacional de Enfermedades Respiratorias Ismael Cosío Villegas, Mexico City 14080, Mexico; 5Departamento de Biologia Celular e Molecular e Bioagentes Patogênicos, Faculdade de Medicina de Ribeirão Preto, Universidade de São Paulo, Ribeirão Preto, São Paulo 14049-900, Brazil

**Keywords:** anti-synthetase syndrome, interstitial lung disease, inflammasome, caspase-1, autoantibodies

## Abstract

Anti-synthetase syndrome (ASSD) is an autoimmune disorder characterized by inflammatory interstitial lung disease (ILD). The main objective of this work was to quantify the concentrations of cytokines and molecules associated with inflammasome activation in bronchoalveolar lavage (BAL) of patients with ASSD and a comparison group of systemic sclerosis (SSc) patients. Cytokines and lactate dehydrogenase (LDH) were determined using the concentrated BAL protein. The activity of caspase-1 and concentration of NLRP3 with the protein purified from the cell pellet in each group of patients. We found higher caspase-1 levels in ASSD vs. SSc, 1.25 RFU vs. 0.75 RFU *p* = 0.003, and LDH levels at 0.15 OD vs. 0.09 OD *p* < 0.001. A significant difference was observed in molecules associated with inflammasome activation, IL-18: 1.42 pg/mL vs. 0.87 pg/mL *p* = 0.02 and IFN-γ: 0.9 pg/mL vs. 0.86 pg/mL, *p* = 0.01. A positive correlation was found between caspase-1 and LDH in the patients with ASSD Rho 0.58 (*p* = 0.008) but not in the SSc group. In patients with ASSD, greater caspase-1 and higher LDH activity were observed in BAL, suggesting cell death due to pyroptosis and activation of the inflammasome pathway.

## 1. Introduction

Anti-synthetase syndrome (ASSD) is a chronic and rare disease that can affect multiple systems across the body. It is considered an immune-mediated disorder characterized by interstitial lung disease (ILD), myositis, arthritis, mechanic’s hands, fever, and Raynaud’s phenomenon [1], along with anti-aminoacyl transfer-RNA-synthetases (anti-tRNA) autoantibodies. Among them, we can include anti-PL12 (anti-alanyl), anti-PL7 (anti-threonyl), anti-EJ (anti-glycyl), anti-OJ (anti-isoleucyl), anti-SC (anti-lysyl), anti-KS (anti-asparaginyl), anti-JS (anti-glutaminyl), anti-Ha (anti-tyrosyl), anti-YRS (anti-threonyl), anti-tryptophanyl and anti-Zo (anti-phenylalanyl), and the most frequent anti-Jo1 (anti-histidyl), being present in 68% of the patients with ASSD [2].

ILD is a manifestation of ASSD associated with high mortality and morbidity; the main tomographic patterns observed are inflammatory, such as organized pneumonia, and cellular (inflammatory) non-specific interstitial pneumonia. In the minority, around 13% of ASSD patients, the ILD manifests as a fibrotic process with a pattern of usual interstitial pneumonia [3].

The association between ASSD and ILD has been previously described [4], and it has been reported that approximately 86% of patients with ASSD develop ILD [5]. One hypothesis for the association between ILD and ASSD is that the alveolar–capillary interface is a dynamic barrier in constant contact with the external environment, even more than the skin, in a continuous cycle of cellular regeneration. Such an environment is equipped with several innate immune sensors that discriminate between self and non-self molecules, surveilling for risks that must be contained or eliminated [6]. Besides that, various genetic antecedents could determine the loss of tolerance due to the presence of neoantigens, which would facilitate the onset of autoimmune phenomena [7]. Furthermore, it has been reported that some antigens can be modified and disseminated by granzyme B activity in lung tissue [8].

One of the innate sensors probably involved in recognizing neoantigens is the inflammasome; a murine model has shown that activating this complex also triggers the cleavage of gasdermin D caspase-1 and caspase-11. The gasdermin D N-terminal fragment oligomerizes at the plasma membrane to form pores, leading to the inflammatory cell death called pyroptosis [9]. The inflammasome is a double-edged sword; its activation is important for controlling infections, but an overactivation may lead to several inflammatory conditions. Various types of inflammasomes have been described as playing a role in several immune-related disorders, such as arthritis, multiple sclerosis, Alzheimer, Parkinson, diabetes, and atherosclerosis [10]. A previous report by our research group showed that the progression of ILD in patients with anti-tRNA autoantibodies is characterized by a high concentration of cytokines related to the Th17 inflammatory response. In addition, IL-1β and IL-18 were two of the elevated cytokines in the progression of ILD in ASSD, suggesting that innate immune sensors play an essential role in the clinical manifestations in ASSD [11]; these two cytokines were described as proteolytic products of caspase-1 after inflammasome activation, a cytosolic multiprotein platform formed after recognizing different stress signals and microbial molecules [12,13]. This study aimed to compare the molecules related to inflammasome activation in patients with ASSD and SSc. To reach this objective, we evaluated the presence of caspase-1, lactate dehydrogenase (LDH), the cytokines IL-1β and IL-18, and the association between them in the bronchoalveolar lavage (BAL) of patients with ASSD and systemic sclerosis (SSc).

## 2. Materials and Methods

### 2.1. Patients, Period, and Treatment

ASSD patients older than 16 years were enrolled between March 2019 and February 2020. Patients were included in this study as ASSD if a diagnosis of ILD was confirmed (TLC < 80% normal FEV1/forced vital capacity (FVC) and FVC < 80%), evidence of ILD in high-resolution chest tomography (HRCT) as well as at least two of the following criteria: idiopathic inflammatory myopathy (IIM) [14] according to Bohan and Peter’s criteria [15,16], fever, arthritis, and mechanic’s hands [17]. Patients have at least one of the following autoantibodies: anti-Jo1, anti-EJ, anti-OJ, anti-PL7, or anti-PL12. We included patients with fibrotic-type ILD secondary to SSc as a comparison group according to the ACR/EULAR 2013 criteria [18]. At baseline, ASSD and SSc patients were evaluated with a BAL before medical treatment for ILD management. The complete description of the recruited patients is presented in Table 1.

We registered the duration of pulmonary symptoms (dyspnea and cough). A complete description of the procedure for pulmonary function tests has been previously described [19]. Baseline serum creatinine kinase (CK) levels and the history of proximal muscle weakness, Raynaud’s phenomenon, sclerodactyly, dermatomyositis rash, and proximal dysphagia modified Borg dyspnea scale and smoking history. The ethics and research committee of the Instituto Nacional de Enfermedades Respiratorias Ismael Cosío Villegas (INER) of Mexico reviewed and approved this protocol, registering it with the code number: B31-20. Informed consent was given to all patients to participate in the study, and those who decided to participate signed the consent, confirming that they knew the objectives and scope of the protocol.

### 2.2. Bronchoalveolar Lavage (BAL) Collection

BAL was collected at the baseline evaluation for the ILD diagnosis and was indicated by the attending physicians as part of the baseline medical evaluation. BAL was performed at the Bronchoscopy Service of the INER with an Olympus 1T180 fiberoptic bronchoscope, under sedation and local anesthesia with 2% xylocaine; in all cases, washing was carried out in the middle lobe or lingula, with instillation of 300 mL of 0.9% NaCl physiological solution at room temperature, in 20 mL aliquots, performing gentle aspiration with plastic syringes of 60 mL, quantifying the recovered volume, average from 260 to 290 mL, which represents 86–96% of the liquid administered. A cell count was carried out in the Department of Morphology of the Research Unit, where it was centrifuged at 300× *g* for 15 min at 4 °C, the supernatant was frozen at −80 °C, and the cell pellet was resuspended in 5 mL of Hank solution. The cell pellet was frozen at −80 °C for later protein purification.

### 2.3. High-Resolution Computed Tomography (HRCT) Evaluation

HRCT was performed at baseline evaluation. Between 20 and 25 CT scan images were acquired for each patient. HRCT was blindly evaluated by two experts (MM and HN M-T) who have a good concordance in assessing the extent of lung disease on HRCT [19]; they classified the tomographic findings according to the official ATS/ERS Statement as a usual interstitial pneumonia (UIP) pattern, an inflammatory non-specific interstitial pneumonia (I-NSIP) pattern, or fibrotic non-specific interstitial pneumonia (F-NSIP) pattern (20-22). Any discrepancy in the interpretation was solved by consensus. The fibrotic component, defined by reticular opacities and inflammation by ground-glass opacities, was graded according to the Goh score (23). The evaluation was used in the analysis of the data. MM has a good intra-observer agreement (intraclass correlation coefficient 0.90; CI: 0.84–0.94) [19].

### 2.4. Autoantibodies

For patients with ASSD, the IgG anti-tRNA (anti-Jo1, anti-PL7, anti-PL12, anti-EJ, and anti-OJ) was measured using EUROIMMUN immunoblot strips (EUROLINE: Myositis Profile 3 (EUROIMMUN AG, Lübeck, Germany) according to the manufacturer’s instructions. In the case of patients with scleroderma, EUROIMMUN immunoblot strips, Euroline systemic sclerosis (nucleoli) IgG, which includes 13 autoantigens, including SCL-70, centromere (A, B), RNA polymerase III (subunits 11 and 155), fibrillarin, NOR90, Th/To, PM/SCL-100, PM/Scl75, Ku, PDGFR and Ro52 (EUROIMMUN AG, Lübeck, Germany). Antibody measurement was performed based on the manufacturer’s instructions. The specificity of the panel is between 98% to 100% for each of the autoantibodies evaluated [20].

### 2.5. Cytokines’ Immunoassays

To quantify the levels of cytokines, BAL was concentrated 50 times using Thermo Scientific Savant RVT5105 Refrigerated Vapor Trap. In this vacuum concentrator, samples are held in a rotor that spins at 1700 RPM creating a centrifugal force, and the protein is dried by evaporation; protein concentration was equated to 200 µg/mL for each sample, and a commercially available ProcartaPlex Human TH1 TH2 CYTO PANEL 11PLEX Th1/Th2 panel (Thermo Fisher Scientific, Waltham, MA, USA, cat. EPX110-10810-901) was used according to the manufacturer’s instructions. Interleukin (IL)-1β, IL-2, IL-4, IL-5, IL-6, IL-12p70, IL-13, IL-18, interferon (IFN)-γ, granulocyte-macrophage colony-stimulating factor (GM-CSF) and tumor necrosis factor (TNF)-α were quantified. Multiplex assays were performed according to the manufacturer’s instructions. Samples were homogenized and adjusted for further quantification in the Luminex^®^LABScan 100 (Luminex Corp. Austin, TX, USA) system. The xPONENT 3.1 software (Luminex Corp. Austin, TX, USA) was used for data acquisition and analysis. Cytokine concentration was calculated from the standard curve using a five-parameter logistic curve fitting, and cytokine levels below the standard range were extrapolated to give approximate values.

### 2.6. Caspase-1 Activity

To determine caspase-1 activity, 50 µg of the purified protein from the BAL cellular pellet was used. The cell pellet obtained from the BAL of both groups was thawed slowly at 4 °C. Subsequently, the protein was purified using Thermo Scientific™ M-PER™ mammalian protein extraction reagent with this highly efficient lysis and extraction reagent compatible with Bradford quantification methodology. According to the manufacturer’s instructions, the activity was detected using a Caspase-1 assay kit (ab39412, Abcam™, Cambridge, UK). All assays were performed in triplicate in three independent experiments.

### 2.7. Enzyme-Linked Immunosorbent Assay (ELISA)

ELISA was used to determine the levels of LDH (ab102526, Abcam™ Cambridge, United Kingdom) and NLRP3 (nucleotide-binding oligomerization domain, leucine-rich repeat and pyrin domain-containing proteins-3) (ab274401, Abcam™, Cambridge, UK) using 50 µg of protein from the previously concentrated BAL according to the manufacturer’s instructions. Duplicated wells were set as samples and standard references. The final absorbance values of proteins were read by a microplate reader at 450 nm, while the means were used to calculate LDH and NLRP3 contents of corresponding samples based on the established standard curves.

### 2.8. Statistical Analysis

Continuous variables are described based on their distribution, with mean and standard deviation, or medians with interquartile range; categorical variables and frequencies are presented in percentages. To compare the distribution of numerical variables, we used the Student’s T-test or the Mann–Whitney U test according to the distribution of the variables. In the case of non-parametric data, the results of the statistical inference were re-evaluated using the Hodges–Lehmann test. Categorical data were compared with Fisher’s exact test. The strength of the association between numeral variables was evaluated with Spearman’s rho. All analyses were 2-sided. The α value was set at 5%. Stata (version 14.2) was used in all analyses.

## 3. Results

### 3.1. Patients

The study recruited nineteen ASSD and seventeen SSc patients (Table 1). In ASSD patients, the most frequent antibody present in the patients from the ASSD group was anti-PL7, followed by anti-OJ (Table 2). In the case of the SSc group, the most common autoantibody was Th/To (12/17 (70.5%)), followed by anti-fibrillarin (5/17 (29.5%)). Table 1 describes the clinical characteristics of both groups. There were no differences in sex, age, and comorbidities when comparing both groups. However, there were expected differences in the frequency of fever and signs of mechanic’s hands in the tomographic patterns and cellularity in the BAL, with increased macrophages in the subjects with SSc and lymphocytes in the ASSD group (Figure 1A,B).

### 3.2. IL-18 and IFN-γ Levels Were Higher in the BAL of ASSD Patients

To explore the inflammatory response involved in both groups, the concentration of cytokines related to the inflammasome activity (IL-1β/IL-18) and several other cytokines IFN-γ, IL-2, IL-4, IL-5, IL-6, IL-12, IL-13, TNF-α, and GMCSF were determined in the BAL. Cytokine quantification showed a higher concentration of IL-1β (statistical tendency *p* = 0.07) and IL-18 (statistically significant, *p* = 0.02), Figure 1C,D; in addition to a higher concentration of IFN-γ (statistically significant *p* = 0.01) (Table 3). IL-1β, the other hallmark of inflammasome activation, was also increased in the group of ASSD patients. However, this difference was not statistically significant. In the other cytokines analyzed, we did not detect or see any difference between the groups; cytokines related to inflammasome activation are shown in Figure 1C,D, and the rest of the cytokine concentrations are shown in Table 3.

### 3.3. LDH Concentration and Caspase-1 Activity Were Higher in the BAL of ASSD Patients

Other ways to detect inflammasome activation are through the levels of LDH release, a characteristic of pyroptotic cell death, and through caspase-1 activity. We used the concentrated protein from the BAL to quantify LDH release and the purified protein from the BAL pellet to measure caspase-1 activity and the levels of NLRP3. Our data demonstrated that caspase-1 activity was also increased in ASSD patients (1.25 RFU (IQR: 1.15–1.35 RFU)) (Figure 1E), compared to the group of SSc patients (0.75 RFU (IQR: 0.62–1.14 RFU)). In agreement with this data, LDH release was higher in the ASSD patient group (1.5 OD (IQR: 1.4–1.9) Vs. 0.09 OD (IQR 0.09–1.0 OD), *p* = 0.0001) (Figure 1F). Finally, the quantification of NLRP3 showed no difference between both groups of patients (Figure 1G).

### 3.4. LDH Was Significantly Correlated with Caspase-1 and IL-18 in ASSD

ASSD patients had a positive correlation between LDH and caspase-1 (*p* = 0.008, rho = 0.58); LDH and IL-18 (*p* = 0.01, rho = 0.55); IL-1β and IL-18 (*p* = 0.006, rho = 0.60); and between IL-18 and IFN-γ (*p* = 0.001, rho = 0.67). While in the group of SSc patients, a significant correlation was only between LDH and IL-18 (*p* = 0.001, rho = 0.72) (Table 4). Finally, we found a significant and statistical correlation between neutrophils and caspase-1 activity in patients with ILD associated with SSc (rho = 0.85, *p* = 0.001, Table 5).

### 3.5. The Extent of Fibrosis Was Higher in SSc Patients

HRCT findings showed no difference in the extent of lung disease; however, it was observed that the extent of fibrosis was significantly higher in the group of SSc patients, while the extent of ground glass was significantly higher in the group of ASSD patients (Figure 2).

### 3.6. NLRP3 Was Significantly Correlated with the Extent of Lung Disease in ASSD Patients

The correlation analysis performed between the extent of lung disease, the extent of fibrosis, and the extent of ground glass with the molecules related to the activation of the inflammasome showed that, surprisingly, only the concentrations of NLRP3 and the extent of lung disease were significantly correlated (rho = 0.53, *p* = 0.0235) in ASSD patients. Furthermore, we found a positive and statistically significant correlation between caspase-1 activity and the presence of neutrophils in BAL (rho = 0.85, *p* = 0.00) (Table 5).

## 4. Discussion

In this work, the main goal was to explore the association between inflammasome activity and anti-synthetase syndrome. We have demonstrated the activation of this mechanism of the innate immune system in this autoimmune entity for the first time.

### 4.1. Autoantibodies

ASSD presents different phenotypes: anti-Jo1 patients have more frequent arthritis and myositis and have a better prognosis than non-anti-Jo1 positive patients, such as anti-PL7 or anti-PL12 ASSD positive patients [21]. In this group of patients, there is an overlap of autoantibodies, a common feature in ASSD; 64% of the patients were positive for anti-PL7/PL12, and only 22% (n = 4) were positive for anti-Jo1. It has been documented that patients with anti-PL7/PL12 have an ILD presentation of up to 90%, much higher than those with anti-Jo1 (with a diagnosis of ILD of around 60%). Similarly, anti-PL7/PL12 patients have greater ILD progression over time [22]. In addition, in patients with a predominance of anti-PL7, increased levels of cytokines from the Th17 pathway and IL-1β and IL-18, molecules related to the inflammasome’s activity, have been described [11].

### 4.2. Scanning for Inflammasome Activation

To explore whether the inflammasome is activated during the development of the ASSD, we determined the cytokines related to its activation in the lung (IL-1β and IL-18), the levels of NLRP3, and the release of LDH, a hallmark of inflammasome activation and pyroptosis.

It has been previously proposed that the activation of the inflammasome is an essential step to induce inflammatory responses in macrophages, cells that act in a dual way, increase phagocytosis, and also initiate tissue healing processes; however, chronic activation of these cells favors fibrotic processes, a significant risk factor for the development of inflammatory/autoimmune diseases [23]. Chronic and uncontrolled inflammasome activation could be critical in initiating and developing rheumatic diseases.

The inflammasome is a molecular platform built of intracellular proteins involved in initiating the inflammatory response triggered by a range of stimuli. The NLRP proteins (NOD-like receptor pyrin domain-containing) are the primary innate sensors capable of initiating activation. An adapter protein, ASC (apoptosis-associated speck-like protein), and procaspase-1 are recruited [24]; this last protein, once activated, cleaves pro-IL-1 and pro-IL-18 into their active and secreted forms [25]. The activation of inflammasomes is due to the union of exogenous molecules that come from bacteria, fungi, or viruses; or to endogenous signals derived from cell damage, such as alarmins, which are released by damaged cells either by cancer or trauma necrosis or ischemia. These molecules are the mediators of the induction of inflammation associated with stress processes [24]. Activation of inflammasomes is reflected clinically by recurrent febrile symptoms and initially localized inflammation, characterized by progressive tissue destruction due to pyroptosis (cell death induced by molecules derived from the activation of inflammasomes) [26]; these are very general findings but regularly present in autoimmune diseases.

### 4.3. Comparison with SSc

We decided to evaluate patients with ILD associated with SSc as a control group because it is an autoimmune entity with a different clinical course than ASSD. In the particular case of the patients enrolled in this protocol as a control group, all patients in the SSc had limited cutaneous disease. SSc with limited cutaneous disease is a slowly progressive disease with fewer inflammatory manifestations than a diffuse cutaneous SSc phenotype [27,28]. Based on our results, we can observe that at least two molecules related to the activity of the inflammasome (LDH and caspase-1, Table 3) and a cytokine product of its activity (IL-18, Table 3 and Figure 1), in addition to a cytokine pro-inflammatory (INF-γ, Table 3) were higher in patients with ILD associated with ASSD. We observed that in the concentration of NLRP3 and IL-1β there was no statistically significant difference in both groups of patients, which would confirm what was previously reported about the activity of the inflammasome present in SSc [29]; this was explored by Diude et al., who established an association between the NLRP1 rs8182352 variant and the development of fibrosis in patients with SSc [30]. Moreover, we found a significant and statistically significant correlation between neutrophils and caspase-1 activity in the group of patients with ILD associated with SSc (rho = 0.85, *p* = 0.001, Table 5), so we cannot rule out the participation of different types of inflammasome in the development of lung lesions; nevertheless, the diverse nature of the autoantigens in each of the cases, as well as the innate sensors that recognize them, could determine the composition of the inflammatory infiltrate in the lung. It has been documented that TLR-4 drives the activation of fibroblasts in SSc [31]. In ASSD, the present report represents the first analysis of an innate mechanism involved in ILD. Various studies suggest an important role for toll-like receptors (TLR) in the initial stages of SSc [32], particularly TLR4 endogenous ligands, including fibronectin, hyaluronan fragments, heat-shock protein (HSP) 70, HSP9, high-mobility group box-1 (HMGB-1), and S100A proteins could engage TLR4 (which is increased in SSc skin and lungs) and synergize with TGF-β to increase fibroblast production [33,34], inducing the pulmonary fibrotic profiles observed in these patients [31] as opposed to ASSD patients, who showed more inflammatory pulmonary profiles in the lungs, with a systemic predominance of inflammation with a Th17 profile, in the patients who progress to ILD [11]. Our results confirm that the inflammasome is not activated in the lung of patients with SSc, at least in a chronic and detectable way; however, activation of this innate mechanism was present in the BALs of ASSD patients.

Our data confirm our previous observation, where it was demonstrated that IL-1β and IL-18 are present in patients characterized by ILD progression. Regarding these cytokines, it has been described that IL-1β and TGF-β are common factors in angiogenesis and the proliferation of the synovial fibroblasts in rheumatoid arthritis [35].

### 4.4. Other Autoimmune Entities and the Possible Role of the Inflammasome in the Progression of ILD

To our knowledge, this report demonstrates the first evidence of inflammasome activation in the lungs of ASSD patients; these findings corroborate other reports from autoimmune diseases such as lupus [36], rheumatoid arthritis [23], and Sjogren´s syndrome [37], in which the inflammasome was described as playing a role. In addition, our data confirm our previous observation, where it was demonstrated that IL-1β and IL-18 are present in patients characterized by ILD progression. It has also been reported that the cytokines associated with inflammasome activation can boost the inflammatory response from both Th1 and Th17 profiles [38,39], suggesting that this immunological mechanism could be involved in the clinical findings at various stages of ASSD development. When we evaluated the levels of LDH release in the patients from both groups, the concentration of LDH in the BAL was increased in patients with ASSD compared to SSc, suggesting an increased inflammasome activation and cell death, which could lead to the epithelial–mesenchymal transition [40]. Indeed, it has been documented that the activity of NLRP3 regulates the polarization towards M2 macrophages through the upregulation of IL-4, which favors chronic inflammatory phenomena and the onset of fibrosis [41]. By analyzing the concentrations of the different molecules related to the inflammasome activity, we found that LDH was significantly associated with caspase-1, NLRP3, and IL-18 in ASSD. Beyond the statistical significance, this finding makes biological sense once LDH is released due to pyroptotic cell death, mediated by a multiprotein complex formed by the other correlated proteins.

On the other hand, in the control group, the production of LDH was low, with no correlation to the other proteins. Finally, the expression of NLRP3 in the BAL was determined, and no difference was observed between the groups. However, this phenotype is instead due to an increased inflammasome activation, rather than the expression of the single molecules. In a model of leishmaniasis in rodents explored by de Carvalho et al., where they compared the concentration of NLRP3 in infected and animals control, they observed no difference in the expression of NLRP3, but there was a larger amount of active inflammasome, demonstrated by the produced cytokines and LDH released [42].

### 4.5. HRCT Findings

The comparison made between the HRCT findings in both groups of patients showed that there were no differences between SSc and ASSD in the extent of lung disease; however, SSc patients were observed to have a greater extent of fibrosis, defined by reticular opacities, while ASSD had a higher inflammation, defined by ground glass opacities. These observations validate our experimental model by comparing two autoimmune entities with pulmonary involvement but with different inflammatory components. Secondly, unexpectedly, the only significant correlation between the molecules related to the activation of the inflammasome and the HRCT findings was between NLRP3 and the extent of lung disease, which strengthens our hypothesis about the participation of this molecule being activated and initiating polymerization of the inflammasome complex.

### 4.6. Therapeutic Perspective

It has been documented that the release of IL-1β induces the production and release of IL-6, which promotes the synthesis of fibrinogen and plasminogen activator inhibitor to increase thrombosis and inhibit fibrinolysis, favoring thrombosis in arteries [43]. Due to the promotion of these effects, IL-1β represents a therapeutic target to be taken into account in the treatment of diseases in which the activation of the inflammasome is demonstrated. A double-blind trial using canakinumab, a monoclonal antibody against IL-1β, demonstrated a beneficial effect in preventing the recurrence of myocardial infarction [44]. On the other hand, anakinra (a selective antagonist of the IL-1 receptor, which inhibits both IL-1α and IL-1β) has shown a significant effect in reducing clinical symptoms in different cardiovascular diseases [44] and cancer [45]. Regarding the use of these therapeutic resources in rheumatological diseases, Ruscitti et al. show that the administration of anakinra had positive effects inhibiting the progression of RA [46]; similar effects have been observed when adding canakinumab to the treatment of patients with juvenile arthritis [47]. There is no history of the use of drugs aimed at controlling the inflammasome in ASSD; however, the demonstration carried out here on its activation in the lung of patients with ILD opens the possibility for future exploration of inhibitory drugs.

### 4.7. Strengths and Limitations of the Study

Among the strengths, considering the fact that ASSD is a rare autoimmune entity, the number of patients enrolled in each group analyzed is significant, and the results are representative. On top of that, the exploration of the inflammasome activity was made directly in the lung, which we consider to be an advantage since it reflects how the inflammatory phenomenon develops in the site described as necessary in the onset and development of ASSD, since serum determinations do not always represent local inflammatory phenomena. The present study has its limitations; perhaps the main one is that there is no histopathological confirmation of the cellular or fibrotic NSIP pattern. Another limitation is that the analysis developed determined the concentrations of cytokines released, which showed some variability, especially IL-1β; however, the determination of several of these molecules related to inflammasome activation opens the possibility of further studies even in the search for therapeutic alternatives.

## 5. Conclusions

Several factors determine the clinical development of ASSD. Even though the predominant anti-antibodies in each patient characterize the different clinical phenotypes, in the present work, we demonstrate that not only elements related to lymphocytes are determinants in this autoimmune entity, but also innate mechanisms such as inflammasome are involved. The molecules derived from these determine the progression of tissue damage and later clinical phenotypes.

## Figures and Tables

**Figure 1 cells-12-00060-f001:**
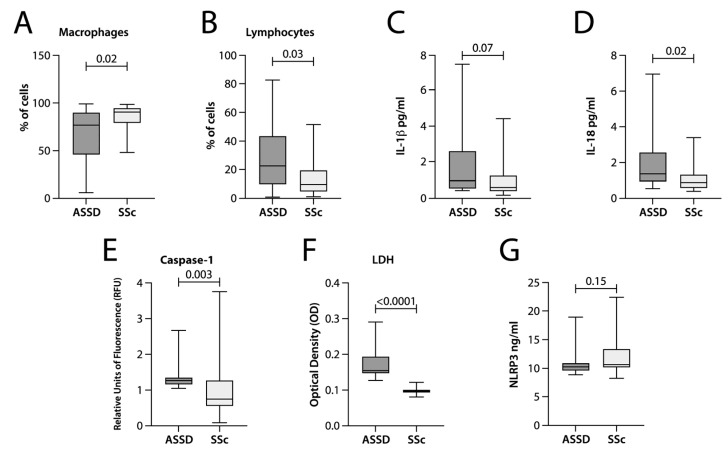
Comparison of molecules related to inflammasome activation quantified in BAL between patients with the anti-synthetase syndrome (ASSD) and systemic sclerosis (SSc). The cellularity was determined by counting with a Neubauer hemocytometer (**A**,**B**); quantifying cytokine concentration in BAL was performed by commercial ProcartaPlex system (**C**,**D**); the caspase-1 activity was quantified by fluorometry (**E**). Finally, the concentration of LDH and NLRP3 was quantified by ELISA (**F**,**G**). Comparisons between groups were made using the Mann–Whitney U test, and the results of the statistical inference were confirmed with the Hodges–Lehmann test.

**Figure 2 cells-12-00060-f002:**
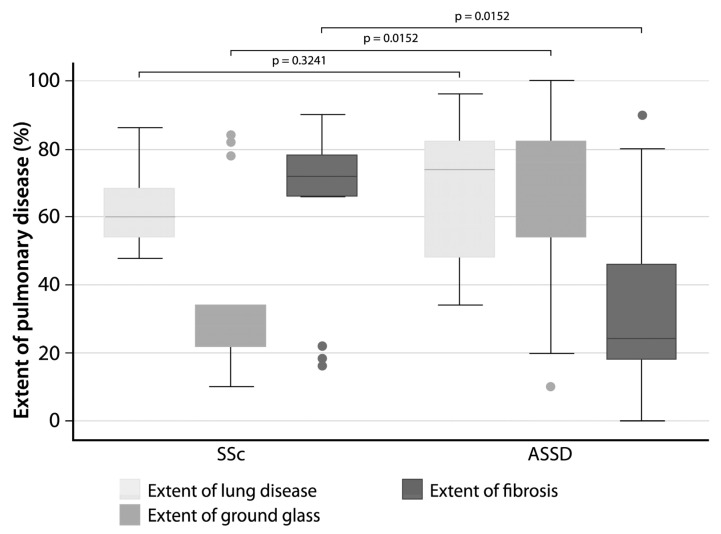
Comparison of tomographic findings between anti-synthetase syndrome (ASSD) patients and systemic sclerosis (SSc). The Goh score in a grayscale shows the total extent of lung disease on HRCT (fibrotic + inflammatory component), the extent of ground glass, and the fibrotic component.

**Table 1 cells-12-00060-t001:** General description of the recruited groups.

Variable	ASSD n = 19	SSc n = 17	*p*-Value
Women *	12 (63%)	12 (70%)	0.7
Age **	63 (52–69)	65 (63–70)	0.15
Arterial * hypertension	6(31%)	5(26%)	0.61
Type 2 diabetes *	4 (22%)	6 (33%)	0.35
Obesity *	5 (26%)	2 (10%)	0.26
PH *	3 (15%)	3 (15%)	0.67
**Clinical characteristics**			
**Fever** *	**5 (27%)**	**0**	**0.047**
Cough *	15 (83%)	15 (83%)	0.73
Dyspnoea *	16 (88%)	15(83%)	0.54
**Mechanic’s Hands** *	**6 (33%)**	**0**	**0.02**
Arthralgias *	6 (33%)	4 (22%)	0.71
Muscular Weakness *	5 (26%)	1(6%)	0.18
Xerostomy *	7 (38%)	7 (38%)	0.92
Dysphagia *	0	4 (23%)	0.02
**Cutaneous sclerosis** *	**0**	**5 (29%)**	**0.01**
Clubbing *	3 (16%)	4 (24%)	0.56
**Bronchoalveolar lavage characteristics**			
**Macrophages (%)** **	**76 (45–90)**	**90 (81–94)**	**0.02**
**Lymphocytes (%)** **	**22 (9.5–43)**	**9.75 (5–18)**	**0.03**
Neutrophils **(%)** **	0 (0–0.5)	0 (0–1)	0.1
Eosinophils **(%)** **	0.25 (0–1.5)	0.25 (0–0.75)	0.7

* Categorical variables are described with percentages. ** Medians (IQR). ASSD: Anti-synthetase syndrome, PH: pulmonary hypertension. Variables in bold indicate statistical differences.

**Table 2 cells-12-00060-t002:** Autoantibodies profile of recruited patients.

Autoantibody	ASSDn = 19	SScn = 17	*p*-Value
Anti-Jo1	4 (22%)	0	0.06
**Anti-OJ**	**6 (33%)**	**0 (0%)**	**0.014**
**Anti-PL7**	**8 (42%)**	**0**	**0.002**
Anti-PL12	4 (22%)	0	0.06
Anti-Ej	3 (11%)	0	0.13
Anti-PM SCL-75	3 (16%)	1 (5.5%)	0.34
Anti-Mi-2	3 (16%)	3 (17%)	0.61
Anti-PM SCL-100	2 (11%)	2 (12%)	0.67
**Anti-TH/To**	**1 (5.5%)**	**12 (70%)**	**0.001**
Anti-Fibrillarin	1 (5.5%)	5 (29%)	0.067
Anti-Ro52	3 (16%)	2 (11%)	0.55
Anti-Nor90	1 (5.5%)	4 (23%)	0.13
Anti-SCL-70	0	2 (11%)	0.21

Categorical variables are described with percentages. Variables in bold indicate statistical differences.

**Table 3 cells-12-00060-t003:** Comparison of cytokines and proteins values of the inflammasome pathway.

Variable	ASSD n = 19	SSc n = 17	*p*-Value
**LDH OD**	**0.15 (0.14–0.19)**	**0.09 (0.09–0.10)**	**0.001**
**Caspase-1 RFU**	**1.25 (1.15–1.35)**	**0.75 (0.62–1.14)**	**0.003**
NLRP3	10.34 (9.76–10.93)	10.69 (10.34–13.02)	0.15
IL-1β pg/mL	1.44 (0.5–4.1)	0.58 (0.45–1.11)	0.07
IL-2 pg/mL	0.75 (0.75–0.79)	0.75 (0.71–0.79)	0.87
IL-4 pg/mL	0.31 (0.26–0.46)	0.29 (0.22–0.36)	0.25
IL-5 pg/mL (n = 13/5)	0.33 (0.13–0.64)	0.34 (0.33–0.44)	0.72
IL-6pg/mL (n =18/15)	7.9 (4.4–21.24)	8.18 (5.77–28.84)	0.61
IL-12 pg/mL	0.41 (0.36–0.45)	0.38 (0.34–0.39)	0.11
IL-13 pg/mL (n = 2/0)	0.14 (0.14–0.31)	-------------------------	-------
**IL-18** pg/mL	**1.42 (0.98–2.65)**	**0.87 (0.62–1.14)**	**0.02**
**IFN-γ**	**0.9 (0.9–1.09)**	**0.86 (0.81–0.95)**	**0.01**
TNF-α pg/mL	0.47 (0.38–0.69)	0.45 (0.38–0.52)	0.50

Medians (IQR). Variables in bold indicate statistical differences. Comparisons between groups were made using the Mann–Whitney U test, and the results of the statistical inference were confirmed with the Hodges–Lehmann test.

**Table 4 cells-12-00060-t004:** Correlation between proteins and molecules related to the inflammasome pathway.

Variable	Both Groups n = 38	ASSD n = 19	SSc n = 17
**Caspase-1/LDH**	**Rho = 0.59** ***p* = 0.0001**	**Rho = 0.58** ***p* = 0.008**	**Rho = 0.15** ***p* = 0.56**
Caspase-1/IL-1β	Rho = 0.43 *p* = 0.007	Rho = 0.34 *p* = 0.14	Rho = 0.28 *p* = 0.25
Caspase-1/IL-18	Rho = 0.69 *p* = 0.001	Rho = 0.19 *p* = 0.42	Rho = 0.40 *p* = 0.10
NLRP3/Caspase-1	Rho = -0.15 *p* = 0.38	Rho = -0.27 *p* = 0.27	Rho = 0.11 *p* = 0.65
**NLRP3/LDH**	**Rho = −0.32** ***p* = 0.05**	**Rho = −0.42** ***p* = 0.07**	**Rho = −0.06** ***p* = 0.81**
NLRP3/IL-1β	Rho = −0.10 *p* = 0.55	Rho = 0.06 *p* = 0.80	Rho = −0.12 *p* = 0.63
NLRP3/IL-18	Rho = −0.17 *p* = 0.30	Rho = −0.29 *p* = 0.23	Rho = 0.15 *p* = 0.56
IL-1β/IL-18	Rho = 0.69 *p* = >0.001	Rho = 0.60 *p* = 0.006	Rho = 0.72 *p* = 0.001
IL-1β/LHD	Rho = 0.36 *p* = 0.02	Rho = 0.26 *p* = 0.26	Rho = 0.15 *p* = 0.55
**IL-18/LHD**	**Rho = 0.55** ***p* = 0.0004**	**Rho = 0.55** ***p* = 0.01**	**Rho = 0.39** ***p* = 0.11**
**IFN-γ/IL-18**	**Rho = 0.57** ***p* = >0.001**	**Rho = 0.67** ***p* = 0.001**	**Rho = 0.24** ***p* = 0.33**

Variables in bold indicate statistical differences.

**Table 5 cells-12-00060-t005:** Correlation between cells of the BAL and molecules related to the inflammasome pathway.

Variable	Both Groups n = 38	ASSD n = 19	SSc n = 17
Caspase-1/macrophages	Rho = 0.36 *p* = 0.83	Rho = 0.10 *p* = 0.66	Rho = 0.39 *p* = 0.12
**Caspase-1/neutrophils**	Rho = 0.12 *p* = 0.48	Rho = −0.3 *p* = 0.87	**Rho = 0.85** ***p* = 0.00**
Caspase-1/lymphocytes	Rho = −0.05 *p* = 0.75	Rho = −0.08 *p* = 0.72	Rho = −0.37 *p* = 0.13
Caspase-1/eosinophils	Rho = −0.01 *p* = 0.91	Rho = −0.11 *p* = 0.63	Rho = −0.26 *p* = 0.30
NLRP3/macrophages	Rho = 0.04 *p* = 0.80	Rho = −0.09 *p* = 0.70	Rho = −0.02 *p* = 0.91
**NLRP3/neutrophils**	Rho = 0.28 *p* = 0.09	**Rho = 0.49** ***p* = 0.03**	**Rho = 0.53** ***p* = 0.02**
NLRP3/lymphocytes	Rho = −0.14 *p* = 0.40	Rho = −0.17 *p* = 0.49	Rho = 0.03 *p* = 0.89
NLRP3/eosinophils	Rho = −0.28 *p* = 0.18	Rho = −0.21 *p* = 0.39	Rho = −0.23 *p* = 0−35
**LDH/macrophages**	**Rho = −0.35** ***p* = 0.03**	Rho = −0.11 *p* = 0.64	Rho = 0.17 *p* = 0.50
LDH/neutrophils	Rho = 0.05 *p* = 0.75	Rho = −0.06 *p* = 0.77	Rho = −0.18 *p* = 0.48
**LDH/lymphocytes**	**Rho = 0.35** ***p* = 0.03**	Rho = 0.16 *p* = 0.51	Rho = −0.14 *p* = 0.58
LDH/eosinophils	Rho = 0.20 *p* = 0.22	Rho = 0.10 *p* = 0.66	**Rho = −0.52** ***p* = 0.02**
IL-1β/macrophages	Rho = −0.17 *p* = 0.32	Rho = −0.08 *p* = 0.72	Rho = 0.24 *p* = 0.33
**IL-1β/neutrophils**	Rho = 0.42 *p* = 0.01	Rho = 0.41 *p* = 0.08	**Rho = 0.11** ***p* = 0.65**
IL-1β/lymphocytes	Rho = −0.01 *p* = 0.92	Rho = −0.13 *p* = 0.58	Rho = −0.24 *p* = 0.33
IL-1β/eosinophils	Rho = −0.02 *p* = 0.88	Rho = −0.08 *p* = 0.72	Rho = 0.03 *p* = 0.89
IL-18/macrophages	Rho = 0.11 *p* = 0.52	Rho = 0.21 *p* = 0.37	Rho = 0.24 *p* = 0.33
**IL-18/neutrophils**	Rho = −0.03 *p* = 0.84	Rho = −0.05 *p* = 0.80	**Rho = 0.57** ***p* = 0.01**
IL-18/lymphocytes	Rho = −0.10 *p* = 0.54	Rho = −0.20 *p* = 0.40	Rho = −0.25 *p* = 0.32
IL-18/eosinophils	Rho = −0.06 *p* = 0.71	Rho = −0.10 *p* = 0.68	Rho = 0.02 *p* = 0.92

Variables in bold indicate statistical differences.

## Data Availability

Authors confirm the raw data to support this study’s conclusions are included in the manuscript. The corresponding author will provide more information, upon reasonable request, to any qualified researcher.

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
