# Peer review of "Enhanced Activity of NLRP3 Inflammasome in the Lung of Patients with Anti-Synthetase Syndrome"

_cells, 2022, doi:10.3390/cells12010060_

Round 1

Reviewer 1 Report

The authors of this paper quantify the concentrations of different cytokines and molecules associated with inflammasome activation in bronchioloalveolar lavage (BAL) of patients with anti-synthetase syndrome (ASSD) and a comparison group of systemic sclerosis (SSc) patients, also correlating them with the extent of lung disease, as shown by High-Resolution Computed Tomography (HRCT) Evaluation. These findings may prove useful for future exploration of inhibitory drugs for the treatment of interstitial lung disease due to autoimmune mechanisms.

The paper offers an informative introduction, which provides the necessary background for the evaluation of the study results. The authors applied appropriate methods and properly interpreted the results. The conclusion is adequately supported by the results. The authors also helpfully list several limitations of the study, which could provide a starting point for further research.

General comments

1.      There are several confusing formulations throughout the paper, therefore I believe that the manuscript would greatly benefit from an English editing service.

2.      The title does not seem completely appropriate, as the paper does not describe a prospective study, as it does not follow and observe a group of subjects over a period of time to gather information and record the development of outcomes.

Specific comments

1.      Page 1, line 3: please remove “the”

2.      Page 1, line 25: “The main of this work” – this is missing a noun, probably “The main objective…”

3.      Page 2, line 71: please specify whether caspase 11 refers to the murin caspase, which has as human homologues caspase-4/5, otherwise it might be construed as a reference to a human caspase

4.      Page 2, line 77: remove “and so on”

5.      Page 3, line 92: there is an open parenthesis that does not close

6.      High-Resolution Computed Tomography (HRCT) Evaluation – this section seems confusing and would probably benefit reformulation. Please specify that MM and HN M-T are the two experts and please reformulate the section as it is extremely similar to your previous paper (Ref. 19).

7.      Page 4, lines 133-135: Please reference the statement “The evaluation of MM was used in the analysis of the data. MM has a good intraobserver agreement (intraclass correlation coefficient 0.90; 134 CI: 0.84 – 0.94).”, as this is copied from Ref. 19, but it is also not referenced there.

8.      Page 5, line 187: “followed by anti-Jo1 (Table 2).” – however, from Table 2 it appears that the second most common autoantibody in ASSD was anti-JO (33%), please check

9.      Page 5, line 188: “followed by anti-NOR-90 (5/17 (29.5%))” - – however, from Table 2 it appears that the second most common autoantibody in SSc was Anti-Fibrillarin (29%), please check

10.   Table 2: per your statement “Variables in bold indicate statistical differences”, therefore anti-OJ autoantibodies, with a p of 0.014, should also be in bold

11.   Page 6, line 215: “statistical trend” is an ambiguous formulation, please replace/reformulate

12.   Page 12, line 307: it is not clear from Table 2 that there was no overlap between the population of patients with anti-PL7 (8 patients) and those with anti-PL12 (4 patients); please provide whether there were any patterns of several present autoantibodies 

Author Response

Reviewer 1

Dear Reviewer:

We appreciate the time dedicated to the new revision of our manuscript, we have endeavored to answer each of the questions posed and we hope that they will find a better work and more suitable for publication.

The authors of this paper quantify the concentrations of different cytokines and molecules associated with inflammasome activation in bronchioloalveolar lavage (BAL) of patients with anti-synthetase syndrome (ASSD) and a comparison group of systemic sclerosis (SSc) patients, also correlating them with the extent of lung disease, as shown by High-Resolution Computed Tomography (HRCT) Evaluation. These findings may prove useful for future exploration of inhibitory drugs for the treatment of interstitial lung disease due to autoimmune mechanisms.

The paper offers an informative introduction, which provides the necessary background for the evaluation of the study results. The authors applied appropriate methods and properly interpreted the results. The conclusion is adequately supported by the results. The authors also helpfully list several limitations of the study, which could provide a starting point for further research.

General comments

  1. There are several confusing formulations throughout the paper, therefore I believe that the manuscript would greatly benefit from an English editing service.

We appreciate the observation and have addressed it by carefully reviewing the writing and grammar of our manuscript.

  1. The title does not seem completely appropriate, as the paper does not describe a prospective study, as it does not follow and observe a group of subjects over a period of time to gather information and record the development of outcomes.

Thanks for the comment. Regarding the title of the manuscript, we have reduced it to avoid confusion by making it clear.

Specific comments

  1. Page 1, line 3: please remove “the”

According to the recommendation, the article was removed from the work's title.

  1. Page 1, line 25: “The main of this work” – this is missing a noun, probably “The main objective…”

We were attending to the indication that the word objective was added in line 25 of the manuscript's abstract.

  1. Page 2, line 71: please specify whether caspase 11 refers to the murin caspase, which has as human homologues caspase-4/5, otherwise it might be construed as a reference to a human caspase

Following the reference in the paragraph, the clarification that the caspase referred to is murine was added.

  1. Page 2, line 77: remove “and so on”

The words mentioned were removed from line 77 of page 2

  1. Page 3, line 92: there is an open parenthesis that does not close

Parentheses were correctly placed on line 92 of page 3

  1. High-Resolution Computed Tomography (HRCT) Evaluation – this section seems confusing and would probably benefit reformulation. Please specify that MM and HN M-T are the two experts and please reformulate the section as it is extremely similar to your previous paper (Ref. 19).

According to the reviewer's recommendation, the HRCT findings paragraph was reformulated.

  1. Page 4, lines 133-135: Please reference the statement “The evaluation of MM was used in the analysis of the data. MM has a good intraobserver agreement (intraclass correlation coefficient 0.90; 134 CI: 0.84 – 0.94).”, as this is copied from Ref. 19, but it is also not referenced there.

We appreciate the recommendation. The paragraph was reformulated, and the reference was added in the appropriate place in the sentence.

  1. Page 5, line 187: “followed by anti-Jo1 (Table 2).” – however, from Table 2 it appears that the second most common autoantibody in ASSD was anti-JO (33%), please check

We appreciate the pointing out. Line 187 on page 5 was corrected in accordance with table 2 of the manuscript.

  1. Page 5, line 188: “followed by anti-NOR-90 (5/17 (29.5%))” - – however, from Table 2 it appears that the second most common autoantibody in SSc was Anti-Fibrillarin (29%), please check

Thanks for the observation; line 188 of page 5 was corrected according to table 2 of the manuscript.

  1. Table 2: per your statement “Variables in bold indicate statistical differences”, therefore anti-OJ autoantibodies, with a p of 0.014, should also be in bold

The line that indicates the OJ antibody is now in bold, indicating the difference between groups in Table 2.

  1. Page 6, line 215: “statistical trend” is an ambiguous formulation, please replace/reformulate

 The sentence was reformulated according to the recommendation

  1. Page 12, line 307: it is not clear from Table 2 that there was no overlap between the population of patients with anti-PL7 (8 patients) and those with anti-PL12 (4 patients); please provide whether there were any patterns of several present autoantibodies 

Clarification of overlapping autoantibodies, a common feature in ASSD, was added to the paragraph.

Reviewer 2 Report

Generally, the results of the study are interesting and the methods used are appropriate. But certain aspects must be clarified:

1. Materials and Methods

The method used for protein purification from the BAL cellular pellet must be specified

How was the bronchioalveolar lavage characterized in both groups (how was the percentage of macrophages, lymphocytes, etc. determined?).

We encourage specifying the LOD for each analyte determined using the ProcartaPlex Human TH1 TH2 CYTO PANEL 11PLEX Th1/Th2 kit panel (Thermo Fisher Scientific, Waltham, MA, USA, cat. EPX110-10810-901).

2. Results

Please provide the raw data for LDH, Caspase1, IL-1β, IL-18, IFN-γ (Table 3) and the applied test, to check the p significance value. The mean values of the two study groups are very close and from our experience it is difficult to be statistically significant.

To argue the extent of fibrosis, we also encourage the evaluation of TGF-β1 and/or IL-8 levels in BAL. IL-1β and IL-6 induce EMT to promote fibrosis via TGF-β signaling and STAT3 signaling. In pulmonary fibrosis, TGF-β signaling induces fibroblast activation and collagen synthesis. Studies have shown that IL-8 contributes to pulmonary fibrosis by targeting macrophages stimulating their migration to the fibroblastic foci.

Is there any significant correlation between IL-1β and IL-6?

Please check the supplementary materials. Supp Table 1 is the same as table 4 from the manuscript, Supp Table 2 is the same as table 5 and the Supp Figure 1 is the same as Figure 2 from the manuscript. In the manuscript, Figure 2 is noted as Figure 1.

3. Discussions

The discussions should be revised, for example the phrase “Regarding these cytokines, it has been described that IL-1β and TGF-β are common factors in angiogenesis and the proliferation of the synovial fibroblasts in rheumatoid arthritis (39) appeared at lines 373-375 as well as 383-385.

Author Response

Reviewer 2

Dear Reviewer:

We appreciate the time dedicated to the new revision of our manuscript, we have endeavored to answer each of the questions posed and we hope that they will find a better work and more suitable for publication.

Generally, the results of the study are interesting and the methods used are appropriate. But certain aspects must be clarified:

  1. Materials and Methods

The method used for protein purification from the BAL cellular pellet must be specified

We appreciate the recommendation. Protein purification methodology in the BAL is now mentioned in the methodology of the manuscript.

How was the bronchioalveolar lavage characterized in both groups (how was the percentage of macrophages, lymphocytes, etc. determined?).

Once the liquid administered for the BAL was obtained, a count was made by an expert technician employing microscope observation by exclusion with Trypan blue as part of the diagnostic procedure for each patient.

We encourage specifying the LOD for each analyte determined using the ProcartaPlex Human TH1 TH2 CYTO PANEL 11PLEX Th1/Th2 kit panel (Thermo Fisher Scientific, Waltham, MA, USA, cat. EPX110-10810-901).

Thank you for the observation; however, the cytokine concentration was calculated from the standard curve using a five-parameter logistic curve fitting; cytokine levels below the standard range were extrapolated to give approximate values.

  1. Results

Please provide the raw data for LDH, Caspase1, IL-1β, IL-18, IFN-γ (Table 3) and the applied test, to check the p significance value. The mean values of the two study groups are very close and from our experience it is difficult to be statistically significant.

Yes, of course, we agree with the complete clarity in evaluating the data of any research project in science. To answer this observation, we present the copy-paste of the results directly from Stata. The data were analyzed according to the distribution of the variables, using the Wilcoxon rank (Mann-Whitney) test or the t-test as appropriate. The reviewer is correct that on many occasions, the lack of significance is not found due to the statistical test's incorrect election according to the variables' distribution.

LDH

. ranksum LDHOD, by(sas)

Two-sample Wilcoxon rank-sum (Mann-Whitney) test

         sas |      obs    rank sum    expected

-------------+---------------------------------

           0 |       17         153       314.5

           1 |       19         513       351.5

-------------+---------------------------------

    combined |       36         666         666

unadjusted variance      995.92

adjustment for ties       -0.64

                     ----------

adjusted variance        995.28

Ho: LDHOD(sas==0) = LDHOD(sas==1)

             z =  -5.119

    Prob > |z| =   0.0000

Caspase 1

. ranksum Caspasa1RelativeCaspase1acti, by(sas)

Two-sample Wilcoxon rank-sum (Mann-Whitney) test

         sas |      obs    rank sum    expected

-------------+---------------------------------

           0 |       17         222       314.5

           1 |       19         444       351.5

-------------+---------------------------------

    combined |       36         666         666

unadjusted variance      995.92

adjustment for ties        0.00

                     ----------

adjusted variance        995.92

Ho: Caspas~i(sas==0) = Caspas~i(sas==1)

             z =  -2.931

    Prob > |z| =   0.0034

IL-1Beta

. ranksum IL1BETApgmL, by( sas)

Two-sample Wilcoxon rank-sum (Mann-Whitney) test

         sas |      obs    rank sum    expected

-------------+---------------------------------

           0 |       17         258       314.5

           1 |       19         408       351.5

-------------+---------------------------------

    combined |       36         666         666

unadjusted variance      995.92

adjustment for ties       -0.38

                     ----------

adjusted variance        995.53

Ho: IL1BET~L(sas==0) = IL1BET~L(sas==1)

             z =  -1.791

    Prob > |z| =   0.0733

IL-18

. ranksum IL18pgmL, by(sas)

Two-sample Wilcoxon rank-sum (Mann-Whitney) test

         sas |      obs    rank sum    expected

-------------+---------------------------------

           0 |       17         242       314.5

           1 |       19         424       351.5

-------------+---------------------------------

    combined |       36         666         666

unadjusted variance      995.92

adjustment for ties       -0.38

                     ----------

adjusted variance        995.53

Ho: IL18pgmL(sas==0) = IL18pgmL(sas==1)

             z =  -2.298

    Prob > |z| =   0.0216

INFγ

. ranksum IFNGpgmL, by(sas)

Two-sample Wilcoxon rank-sum (Mann-Whitney) test

         sas |      obs    rank sum    expected

-------------+---------------------------------

           0 |       17       235.5       314.5

           1 |       19       430.5       351.5

-------------+---------------------------------

    combined |       36         666         666

unadjusted variance      995.92

adjustment for ties      -12.05

                     ----------

adjusted variance        983.87

Ho: IFNGpgmL(sas==0) = IFNGpgmL(sas==1)

             z =  -2.519

    Prob > |z| =   0.0118

Attached to this response document, the reviewer will find an excel file with the raw data of the requested variables. The data are coded to identify the SAS variables with 1 as a patient with ASSD and 0 as a control group patient with SSc.

To argue the extent of fibrosis, we also encourage the evaluation of TGF-β1 and/or IL-8 levels in BAL. IL-1β and IL-6 induce EMT to promote fibrosis via TGF-β signaling and STAT3 signaling. In pulmonary fibrosis, TGF-β signaling induces fibroblast activation and collagen synthesis. Studies have shown that IL-8 contributes to pulmonary fibrosis by targeting macrophages stimulating their migration to the fibroblastic foci.

We appreciate the recommendation regarding the fibrotic pathway and the cytokines to be explored. Unfortunately, the biological material obtained from the patients in this protocol was used in its entirety for the reported determinations; of course, we have planned a continuation of this work for the near future, in which we will explore the fibrotic pathway induced by these signals produced in the lung with a focus on diagnosis, prognosis, and treatment of the antisynthetase syndrome.

Is there any significant correlation between IL-1β and IL-6?

Thanks for the question; after the correlation analysis, we did not find a correlation between IL1 beta and IL-6 in this work.

Please check the supplementary materials. Supp Table 1 is the same as table 4 from the manuscript, Supp Table 2 is the same as table 5 and the Supp Figure 1 is the same as Figure 2 from the manuscript. In the manuscript, Figure 2 is noted as Figure 1.

Thanks for the observation. The supplementary materials were incorporated into the main text at the editor's recommendation.

  1. Discussions

The discussions should be revised, for example the phrase “Regarding these cytokines, it has been described that IL-1β and TGF-β are common factors in angiogenesis and the proliferation of the synovial fibroblasts in rheumatoid arthritis (39) appeared at lines 373-375 as well as 383-385.

Thanks for the observation, the discussion was revised, and the same sentence was corrected.

Round 2

Reviewer 2 Report

The Mann-Whitney test compares differences between distributions, not medians. I checked the raw data and indeed the "p" value shown in table 3 is correct, but it represents the differences between the distributions of the 2 groups of patients, although the table showed the medians. Applying Independent-samples test Hodges Lehmann for median difference, statistically significant differences between medians are detected in the case of LDH and Caspase 1. Consequently, please specify in Figure 1 and table 3 what represent the p values and and make the changes in the manuscript accordingly.

Author Response

Dear reviewer:

We appreciate the attention devoted to this new revision of our manuscript, we have endeavored to respond to your comment satisfactorily, and we hope that you will find a better and more suitable work for publication.

The Mann-Whitney test compares differences between distributions, not medians. I checked the raw data and indeed the "p" value shown in table 3 is correct, but it represents the differences between the distributions of the 2 groups of patients, although the table showed the medians. Applying Independent-samples test Hodges Lehmann for median difference, statistically significant differences between medians are detected in the case of LDH and Caspase 1. Consequently, please specify in Figure 1 and table 3 what represent the p values and and make the changes in the manuscript accordingly.

R. We appreciate the observation of our work, this observation allows us to present the results of our research more clearly; With the suggested test it was possible to establish the agreement between the differences and the p values. The materials and methods section was corrected; and the figure captions and tables indicate how the comparisons were made